# Influence of the Photodegradation of Azathioprine on DNA and Cells

**DOI:** 10.3390/ijms232214438

**Published:** 2022-11-20

**Authors:** Mihaela-Cristina Bunea, Victor-Constantin Diculescu, Monica Enculescu, Daniela Oprea, Teodor Adrian Enache

**Affiliations:** 1National Institute of Materials Physics, Atomistilor 405A, 077125 Magurele, Romania; 2Faculty of Physics, University of Bucharest, Atomistilor 405, 077125 Magurele, Romania

**Keywords:** azathioprine, photodegradation, spectrophotometry, voltammetry, L929

## Abstract

Azathioprine (AZA) is a pharmacologic immunosuppressive agent administrated in various conditions such as autoimmune disease or to prevent the rejection of organ transplantation. The mechanism of action is based on its biologically active metabolite 6-mercaptopurine (6-MP), which is converted, among others, into thioguanine nucleotides capable of incorporating into replicating DNA, which may act as a strong UV chromophore and trigger DNA oxidation. The interaction between azathioprine and DNA, before and after exposure to solar simulator radiation, was investigated using UV–vis spectrometry and differential pulse voltammetry at a glassy carbon electrode. The results indicated that the interaction of AZA with UV radiation was pH-dependent and occurred with the formation of several metabolites, which induced oxidative damage in DNA, and the formation of DNA-metabolite adducts. Moreover, the viability assays obtained for the L929 cell culture showed that both azathioprine and degraded azathioprine induced a decrease in cell proliferation.

## 1. Introduction

Azathioprine, one of the oldest pharmacologic immunosuppressive agents, was developed in 1957 by the scientists Gertrude Elion and George Hitchings who received, and shared with James W. Black, the Nobel prize in Physiology or Medicine in 1988 for their discoveries of important principles for drug treatment [1,2,3]. The development of azathioprine as a purine-mimic antimetabolite represented the successful results of the therapeutic improvement of 6-mercaptopurine (6-MP) through sulfur-substituted compounds [4]. Azathioprine became the medication used to prevent the rejection of transplanted organs as well as for the treatment of autoimmune diseases [5,6,7,8,9]. It presents a high adsorption rate from the gastrointestinal tract, the serum half-life is up to 0.5 h, the biologic half-life is approximately 24 h, and 45% of the drug is excreted in the urine, the remaining portion being metabolized to 6-MP in red blood cells through the action of glutathione [10]. Subsequently, 6-MP is enzymatically converted into 6-TUA, 6-MMP, and 6-TGNs, see Figure 1. 6-TGNs are thioguanine nucleotides containing different numbers of phosphate groups, which are capable of incorporating into replicating DNA, a process which blocks the de novo pathway of purine synthesis and inhibits DNA, RNA, and protein synthesis [3,11].

Although it presents a more favorable therapeutic index than 6-MP, the use of AZA is associated with serious adverse effects, and the susceptibility to toxicity varies with several factors such as age, genetics, and dosage [12,13]. Generally, the adverse effects include gastrointestinal symptoms, bradycardia, hepatotoxicity, and myelosuppression [14]. However, the highest concern is represented by the fact that the incorporation of AZA metabolites into the DNA increases skin sensitivity to the Sun’s incident ultraviolet (UVA) radiation and, consequently, the skin cancer incidence [13,15,16,17]. Different to endogenous DNA, which has a low UV adsorption degree, DNA sequences containing AZA metabolites act as strong UV chromophores, which, through different pathways, can trigger DNA oxidation, breakdown, crosslinking, and covalent attachment to proteins [16,18,19,20,21].

The UV degradation of azathioprine and its metabolites was investigated in numerous scientific papers due to the association between the use of this immunossuppresive drug and skin cancer [22]. Smaranda et al. revealed that the photodegradation of azathioprine is induced in the presence of UV light and the O_2_ in air [23]. Zhang et al. revealed that the degradation rate of AZA increased with an increase in the content of oxidants [2].

The high risk of skin cancer in patients using azathioprine has led to development of numerous spectrophotometric studies relating to the effects of the UV damage of AZA and its metabolites [18,19,20,23,24]. The novelty of this study consisted both in highlighting the interaction between UV degradation products and DNA through electrochemical methods, before and after photodegradation, and in studying their effects on the fibroblast cells L929, with the expectation that the results obtained will bring new insights on the influence of the photodegradation of azathioprine on DNA and cells. The effects of the photodegradation of AZA on DNA and the fibroblast cells L929 i was followed using UV–vis spectrophotometry and differential pulse voltammetry, while the interaction of AZA, before and after photodegradation, with an L929 cell culture was investigated by the evaluation of cell viability and using fluorescence microscopy.

## 2. Results and Discussion

### 2.1. Spectrophotometric Characterization

The samples of AZA, IMI, or MP were analyzed by UV spectrophotometry in the absence and in the presence of dsDNA in 0.1 M pH 7.0 phosphate buffer before and after exposure to the solar simulator (SS) for different lengths of time.

The UV spectra of the fresh dsDNA sample showed the typical absorption band at 260 nm, as shown in Figure 1. Exposure to the solar simulator for different times led to a small hyperchromic effect of an approx. 1% increase in the initial absorbance values, which was related to the unwinding of the double helical structure upon UV irradiation. A control experiment of a dsDNA solution without exposure to the solar simulator did not show any variation in the absorbance value in this time interval.

The UV spectra of the AZA samples were also recorded after different of exposure times to the SS, Figure 2A. The spectra recorded at a 0 min exposure time depicted a maximum at 280 nm, which was assigned to the imidazol moiety of AZA [25]. Upon exposure to the SS, a decrease in this absorption band with an increasing exposure time was observed. At the same time, a newly well-defined absorption band occurred at 334 nm, which indicated the cleavage of azathioprine, which was in agreement with the mechanism in Figure 1 and was simultaneous with the generation of 6-mercaptopurine and 1-methyl-4-nitrol-5-imidazol [17].

The UV spectra of the AZA-dsDNA samples were recorded after different times of exposure to the SS, as shown in Figure 2B. The spectra at a 0 min incubation time showed both the DNA and AZA absorption bands, although a blue shift (approx. 3 nm) was observed for the AZA band, proving that an interaction between the dsDNA and AZA occurred through the imidazole moiety. With increasing the exposure time to the SS, a continuous decrease in the absorbance of AZA at 277 nm occurred accompanied by the appearance of a band at 334 nm due to the cleavage of AZA. Nonetheless, the DNA absorption band practically maintained the same absorbance value and position, as shown in Table 1.

Since AZA degrades under UV radiation and undergoes a cleavage to MP and IMI derivatives, a detailed investigation of these two compounds and their interaction with dsDNA was also carried out, as shown in Figure 3.

The UV spectra of the MP were recorded after different times of exposure to the solar simulator, as shown in Figure 3(A1). At 0 min, a maximum at 322 nm was observed, which progressively decreased with increasing the exposure time. At the same time, a new absorption band occurred at 274 nm, which was attributed to the MP degradation product. The interaction of MP with dsDNA was also investigated during exposure to the SS, as shown in Figure 3(A2). The spectrum recorded at a 0 min exposure time showed a clear hyperchromic effect at 260 nm, as shown in Table 1, which indicated a conformational modification within the DNA structure. By increasing the irradiation time, the MP adsorption band at 322 nm decreased and the increase at 260 nm was due to both the DNA and MP degradation products.

Similarly, the UV spectra of the IMI were recorded after different times of exposure to the SS, Figure 3(B1). The IMI showed an adsorption band at 299 nm, which decreased when increasing the exposure time to the SS, thus leading to an IMI degradation product, which was identified through the absorption maximum at 356 nm. The spectrum recorded for the IMI interaction with dsDNA did not show any relevant modification within the dsDNA conformation, as shown in Figure 3(B2) and Table 1.

Spectrophotometric investigations of the interactions of DNA with chemical compounds are usually followed by monitoring the DNA absorption band at 260 nm as well as the absorption bands of the analytes. However, the absorption band of DNA does not provide any specificity for the DNA bases. On the contrary, electrochemical methods allow the monitoring of all DNA bases, since their oxidation occurs at different potential values. Hence, in order to gain insights into the mechanism of the interaction of DNA with the photodegradation products of AZA, electrochemical experiments were also performed following a similar procedure to the one used for the spectrophotometric investigations.

### 2.2. Electrochemical Characterization

Differential pulse voltammograms were recorded before and after the UV irradiation of the individual solutions of either dsDNA or AZA, IMI, or MP. In order to probe some of the potential damages and conformational modifications produced by the UV degradation of AZA, IMI, or MP to the dsDNA, DP voltammograms were recorded in the presence and in the absence of the AZA immunosuppressive drug and these metabolites by either monitoring the oxidation currents of the purine bases guanine and adenine within the dsDNA helix or the occurrence of new peaks related to free bases or their oxidation products such as 8-oxoguanine and 2,8-dihydroxiadenine, which typically occur at lower potential values.

Initial measurements were recorded for the dsDNA samples before and after exposure to the solar simulator, as shown in Figure 4. The DP voltammograms showed the oxidation peaks of guanine at Epa = +0.89 V and adenine at Epa = +1.15 V. An increase in the peak currents was identified after sample irradiation [19], and this effect was associated with the conformational modifications the double helix, which led to the exposure of the bases at the electrode surface, facilitating their oxidation [20]. Control experiments were also performed, and a solution of dsDNA was held in the same conditions without exposure to the solar simulator; however, no modification of the peak currents was observed during the timeframe of the experiment.

The electrochemical behavior of AZA was also investigated before and after irradiation, as shown in Figure 5A. The DP voltammogram recorded in the anodic potential range before AZA solution irradiation did not reveal any oxidation process. After exposure to the solar simulator for 180 min, two anodic charge transfer reactions at E_AZA1ss_ = +0.80 V and E_AZA2ss_ = +1.10 V corresponding to AZA degradation products were observed.

The effect of AZA degradation on DNA was also investigated in incubated solutions, as shown in Figure 5B. The DP voltammogram of the fresh AZA-dsDNA sample in a positive potential range revealed the typical oxidation peaks of desoxyguanosine (dGuo) and desoxyadenosine (dAdo) at E_dGuo_ = +0.88 V and E_dAdo_ = +1.15 V, demonstrating that the immunosuppressive drug AZA did not induce a modification in the electrochemical response of the DNA. Upon irradiation, the voltammogram showed both peaks corresponding to the AZA degradation products, D_AZA1_ and D_AZA2_, as well as those corresponding to DNA oxidation, dGuo and dAdo. Nonetheless, the modification of the dsDNA peak currents and the potential values observed on the voltammogram recorded for the irradiated AZA-dsDNA sample, when compared to the standard dsDNA irradiated solution, indicated that a conformation/structural modification of dsDNA occurred during the AZA degradation. A decrease in the dGuo current and a potential shift to more positive values was observed. Regarding the dAdo oxidation potential, a similar potential shift was also observed, but the current increased. Taking into consideration previous investigations on the interaction mechanisms of AZA and its redox products with dsDNA [21], the behavior described above was explained considering the formation of adducts between the guanine residues of DNA and the AZA metabolites in the reaction that impeded their oxidation, which explained the decrease in the dGuo peak. At the same time, this interaction mechanism induced a distortion of the dsDNA double helix, exposing the bases to the electrode surface and consequently increasing in their oxidation currents, as observed for the dAdo peak current.

In order to gain insights into the interaction mechanisms of DNA with UV-irradiated AZA, similar experiments were performed with the AZA degradation products 6-mercaptopurine (MP) and 1-methyl-4 nitro-imidazole (IMI), as shown in Figure 6.

For 6-mercaptopurine, the DP voltammogram recorded in the anodic potential range before irradiation showed one peak at E_MP_ = +0.88 V, which disappeared after exposure to the solar simulator, while a new anodic signal occurred at E_MPss_ = +0.99 V, as shown in Figure 6(A1). The effect of MP degradation on DNA was investigated, and the voltammogram recorded after irradiation showed, on the one hand, a peak characteristic to the MP degradation product. On the other hand, the dAdo oxidation peak increased, and dGuo maintained the same current when compared to the voltammogram recorded in the same solution before irradiation, as shown in Figure 6(A2). This effect could be associated with a preferential interaction between the MP degradation product and the adenine reach segments in the DNA structure, leading to conformational modifications and the exposure of adenine residues to the electrodes surface, thus facilitating their oxidation.

For 1-methyl-4 nitro-imidazole, the DP voltammogram obtained before irradiation did not show any oxidation peaks, yet exposure to the solar simulator led to an oxidation signal at E_IMIss_= +0.77 V, as shown in Figure 6(B1). Nonetheless, the degradation of IMI in the presence of DNA showed a decrease in the dGuo and an increase in the dAdo peaks, similar to the experiments carried out in the presence of AZA, as shown in Figure 6(B2). This proved that the degradation product of IMI was responsible for the adduct formation with guanine residues.

### 2.3. Cell Viability and Proliferation

The viability of L929 cells was evaluated using an MTS assay and fluorescence microscopy on cells cultured and treated with AZA, IMI, and MP solutions prepared before and after 180 min exposure to the solar simulator. The concentrations used were 50 and 100 µM for each compound.

The MTS assay results obtained for cells incubated with AZA solutions, before and after SS photodegradation, indicated a slight decrease in cell viability, compared to the control, of about 20 percent for the concentration of 100 µM. In contrast with the spectrophotometric and electrochemical results, the viability decrease of ~20% did not indicate cytotoxicity effects, and this could be explained by the fact that, differently to the DNA in the solution, the cell’s DNA was protected and repaired by cellular mechanisms.

The decrease cell viability was also observed for the cells incubated with IMI and MP, and the lowest viability values were obtained for the cells incubated with an MP solution, as shown in Figure 7A. However, the viability remained around 80%, meaning that, at the experimental conditions, the AZA, IMI, MP, and their degradations products did not affect the fibroblast L929.

The fluorescence images obtained for the L929 cell culture incubated with AZA, IMI, and MP before and after their exposure to the solar simulator revealed that all the cells presented a well-defined nucleus, and the only difference observed between the samples was the number of viable cells, as shown in Figure 7B–H. Compared with the control experiments, as shown in Figure 7B, the number of viable cells decreased slightly after incubation with AZA, IMI, and MP, both for the fresh solutions and for those exposed to the solar simulator, as shown in Figure 7C–H, and the lowest density of viable cells was obtained for the AZA and MP solutions exposed to the solar simulator, as shown in Figure 7F,H, which was in agreement with the results obtained by the MTS assay, as shown in Figure 7A.

## 3. Materials and Methods

### 3.1. Materials

Azathioprine (AZA), 1-methyl-4 nitro-imidazole (IMI), and 6-mercaptopurine (MP), double strand DNA (dsDNA-catalog number D1501), were acquired from Sigma-Aldrich. All the substances were used without further purification.

Acetate (pH = 4.5) and phosphate buffer (pH = 7.0) electrolyte solutions were prepared with analytical grade reagents and purified water from a Milli-Q system (conductivity below 0.1 μS cm^−1^).

The stock solutions of 4 mM for AZA, IMI, and MP in DMSO and 100 μg mL^−1^ dsDNA in water were prepared daily, kept at 4 °C, and protected from light. Solutions of different concentrations were prepared by dilution in a desired buffer.

All experiments were conducted at room temperature (25 ± 1 °C).

### 3.2. Instrumentation

#### 3.2.1. Solar Simulator Irradiation

The samples subjected to a solar simulator (SS) were irradiated in the spectral range of the highest intensity between 300–800 nm using an SF300 Small Collimated Beam Solar Simulator (Sciencetech, London, ON, Canada) equipped with an Air Mass AM1.5G Filter (spot size: 25 mm diameter at one Sun) and an integrated electrical shutter with a controller and a Xe lamp (300 W). The sample was positioned at 10 cm from the source.

#### 3.2.2. UV-Vis Spectrophotometry

UV–vis spectra were recorded using a UV–Vis–NIR CARY 5000 (Varian, Agilent Technologies Deutschland GmbH, Waldbronn, Germany) spectrophotometer provided with a quartz cell with a light path of 10 mm. The optical absorbance was measured at different time intervals in the λ = 200–800 nm spectral domain. During the optical measurements, all solutions were kept in the dark. All the experiments were carried out by keeping the concentration of AZA, IMI and MP (2 µM) and DNA (10 mg/mL) constant.

For the MTS assay, the absorbance values at 490 nm were recorded using a plate reader FLUOstar Omega (BMG Labtech, Ortenberg, Germany).

#### 3.2.3. Voltammetric Parameters and Electrochemical Cells

The differential pulse (DP) voltammograms were recorded using a computer-controlled Ivium potentiostat with IviumSoft version 2.219 (Ivium Technologies, Eindhoven, The Netherlands). Measurements were carried out using a glassy carbon (GC) working electrode (d = 1.6 mm), a Pt wire counter, and Ag/AgCl (3 M KCl) as a reference electrode in a one-compartment 2 mL electrochemical cell. Before each experiment, the GC electrode was polished using a diamond spray (particle size 1 μm) on a microcloth pad, rinsed with Milli-Q water, and electrochemically pre-treated by recording various DP voltammograms in buffer supporting electrolyte until a steady state baseline voltammogram was obtained. All measurements were performed in N_2_-saturated solutions in the dark in order to prevent the degradation of the immunosuppressive drugs.

DP voltammograms were recorded with a pulse amplitude of 50 mV, pulse width of 100 ms, and a scan rate of 5 mV s^−1^. All the voltammograms presented were background-subtracted and baseline-corrected using the IVIUM soft program tools. This mathematical treatment reduced the peak heights by up to 10%, and it was used in the presentation of all experimental voltammograms for a better and clearer identification of the peaks. The values for peak current presented in all graphs were determined from the original untreated voltammograms.

#### 3.2.4. Fluorescence Microscopy

The fluorescence microscopy images were obtained using a Leica DM6B upright fluorescence microscope (Leica Microsystems CMS GmbH, Wetzlar, Germany) equipped with a Leica CTR6 LED (electronic box containing the power supply for the electronics and the lamps) and a Leica EL6000 external light source for fluorescence excitation. The samples were imaged using a 40× objective (0.65 NA, 0.36 mm WD, and a correction ring) from Leica, an appropriate filter cube (excitation filter 480/50 nm, dichroic mirror 505–510 nm, and emission filter 527/30 nm), and a 4.2 MP sCMOS Leica DFC9000 monochrome fluorescence camera.

### 3.3. Procedures

#### 3.3.1. Photodegradation

For each sample, a volume of 3 mL containing 2 µM AZA, IMI, or MP and 10 mg/mL DNA was exposed to the solar simulator for a period of time between 5 and 180 min at one Sun. The samples were positioned at 10 cm from the source.

#### 3.3.2. Viability Assay

For the in vitro studies, the fibroblast L929 cell line from ATCC (Manassas, VA, USA) was used. Other materials were Dulbecco’s Modified Eagle Medium (DMEM) and an MTS assay kit from Sigma-Aldrich, phosphate buffer saline (PBS) from Thermo Fisher Scientific (Waltham, MA, USA), fetal bovine serum (FBS) from Thermo Fisher Scientific, trypsin from Lonza Bioscience Solutions (Basel, Switzerland), and antibiotics from Biological Industries (Kibbutz Beit-Haemek, Israel). The cells were grown in Dulbecco’s Modified Eagle Medium (DMEM) culture medium supplemented with 4.5 g/L glucose, 2 mM l-glutamine, 10% fetal bovine serum, penicillin (100 U/mL), and streptomycin (100 μg/mL) under controlled conditions (humidity, 5% CO_2_, 37 °C). Sub-cultivation was conducted in cell culture flasks (T-25), and when a pre-confluence of ~ 80% was reached, the cells were detached using trypsin solution of 0.25% concentration, counted, and plated for the experimental procedures. For the viability assay, the cells were seeded in 96-well plates at a density of 7000 cells/well and placed in an incubator at 37 °C, 5% CO_2_. After 48 h incubation, the medium was changed.

AZA, IMI, and MP solutions were added at two concentrations, 50 and 100 µM, and incubated for 3 h. Following incubation, the medium was changed, and 10 µL MTS solution was added to each well, and after 4 h the absorption at 490 nm was recorded using a plate reader FLUOstar Omega, BMG Labtech, Germany.

#### 3.3.3. Fluorescence Microscopy

Fixation of the cells was performed by incubating the cells for 15 min at room temperature using a solution of 3% paraformaldehyde and 0.2% glutaraldehyde in PBS. Then, the cells were washed with PBS and stained with a solution of 2 mg/mL Acridine-Orange in the dark for the 20 min. Finally, the cells were washed with PBS and mounted on a glass slide and imaged using the microscope.

## 4. Conclusions

Azathioprine is a pharmacologic immunosuppressive agent, which, in specific conditions, presents dangerous side effects including carcinogenesis, which are most likely due to its photosensitivity. The degradation of azathioprine leads to 1-methyl-4 nitro-imidazole and 6-mercaptopurine metabolites, which, upon their conversion, involve other degradation products. The interaction of azathioprine and its degradation products with dsDNA was investigated before and after exposure to a solar simulator by UV–vis spectrophotometry and voltammetry. The interaction of AZA with dsDNA led to the formation of adducts between the guanine residues of DNA and AZA metabolites, inducing a distortion of the dsDNA double helix, exposing the bases to the electrode surface, and consequently causing an increase in their oxidation currents. Insights into the interaction mechanisms were obtained by an investigation of the interaction of dsDNA with the azathioprine degradation products 1-methyl-4 nitro-imidazole and 6-mercaptopurine. It was shown that 1-methyl-4 nitro-imidazole degradation was responsible for adduct formation with guanine residues, while 6-mercaptopurine metabolites interacted preferentially with adenine reach segments, inducing a distortion of the DNA strands. The in vitro assays using the fibroblasts L929 showed a slight decrease in cell viability in the presence of AZA and its photodegradation products, and the lowest viability was obtained for the AZA and MP after interaction with solar simulator radiation. However, the decrease was around 20%, meaning that the cells were not affected during the incubation time.

## Data Availability

All data are available on request.

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
