# Peer review of "Influence of the Photodegradation of Azathioprine on DNA and Cells"

_ijms, 2022, doi:10.3390/ijms232214438_

Round 1
Reviewer 1 Report
Manuscript (ijms-1981214) entitled Influence of photodegradation of azathioprine on DNA and cells is acceptable for publication in International Journal of Molecular Sciences.
Author Response
- Manuscript (ijms-1981214) entitled Influence of photodegradation of azathioprine on DNA and cells is acceptable for publication in International Journal of Molecular Sciences.
Thank you for your decision.
Reviewer 2 Report
The conclusion for in vitro cytotoxicity test "The in vitro assays using L929 showed a decrease of cell viability in the presence of AZA and theirs photodegradation products and the lowest viability was obtained for AZA and MP after interaction with solar simulator radiation." is invalid. There is no significant statistic difference could be observed as the data presented in Figure 7A. The author should refine this experiment to improve the reliability of the results.
The author reported azathioprine metabolites after light irradiation induced oxidative damage in DNA and formation of DNA-metabolites adducts. The topic is interesting and the experiments in the cuvette confirm the hypothesis. However, the cytotoxicity test result showed no significant difference between the light treated and dark treated samples.
The author should concerned the comments listed below: There are some spelling mistakes that need to be revised. In 2.3.1. Photodegradation,there should be a space between the number and the unit “3mL 10mg/mL”. Similar mistake could be found in 2.3.2. Viability assay:CO2, 48h and 3h; 2.2.3. N2
For the figure caption of Fig. 7, the author should added more information about the dye used and the excitation and emission wavelength range.
In Scheme 1, the chemical structure of 6-MP should be regenerated. I would suggest the author out all the chemical structure of all the chemical structure involved metabolism of azathioprine.
For the cytotoxicity test, there is no significant different cell viability between the dark and solar treatment. The AO staining assay also showed similar results which is hard to verify the different cytotoxicity. Since the UV-Vis spectra of the AZA, IMI and MP change significantly after light treatment, the author should discuss the reason that result in the inconspicuous cytotoxicity.
Author Response
- The conclusion for in vitro cytotoxicity test "The in vitro assays using L929 showed a decrease of cell viability in the presence of AZA and theirs photodegradation products and the lowest viability was obtained for AZA and MP after interaction with solar simulator radiation." is invalid. There is no significant statistic difference could be observed as the data presented in Figure 7A. The author should refine this experiment to improve the reliability of the results.
The author reported azathioprine metabolites after light irradiation induced oxidative damage in DNA and formation of DNA-metabolites adducts. The topic is interesting and the experiments in the cuvette confirm the hypothesis. However, the cytotoxicity test result showed no significant difference between the light treated and dark treated samples.
We agree with the reviewer comment. Indeed, the viability decrease of ~ 20% do not indicate the cytotoxicity. An explanation for relative good viability, in contrast with the spectrophotometric and electrochemical results, is the fact that different than DNA in solution the cells present protective mechanisms as well as repairing ones. Therefore, the main conclusion of the viability studies is the fact that the cells are not affected by AZA or its degradation products for short incubation periods. However, in a future study it will be interesting to evaluate the behavior of L929 cell culture in the presence of AZA for a long period of time.
- The author should concerned the comments listed below: There are some spelling mistakes that need to be revised. In 2.3.1. Photodegradation,there should be a space between the number and the unit “3mL 10mg/mL”. Similar mistake could be found in 2.3.2. Viability assay:CO2, 48h and 3h; 2.2.3. N2.
Thank you for your suggestions. The text was revised and corrected.
- For the figure caption of Fig. 7, the author should added more information about the dye used and the excitation and emission wavelength range.
The figure capture was revised.
- In Scheme 1, the chemical structure of 6-MP should be regenerated. I would suggest the author out all the chemical structure of all the chemical structure involved metabolism of azathioprine.
The scheme was modified and inserted in the manuscript.
- For the cytotoxicity test, there is no significant different cell viability between the dark and solar treatment. The AO staining assay also showed similar results which is hard to verify the different cytotoxicity. Since the UV-Vis spectra of the AZA, IMI and MP change significantly after light treatment, the author should discuss the reason that result in the inconspicuous cytotoxicity.
The manuscript was revised. The results obtained by fluorescence microscopy are in good agreement with those obtained using the MTS assay.
Reviewer 3 Report
This article focuses on the photodegradation of azathioprine (AZA), an immunosuppressive agent, and its interaction with DNA through thioguanine, resulting from the conversion of its biological metabolite 6-mercaptopurine. The interaction of AZA and DNA, before and after exposure at solar simulator radiation, was investigated by two different methods, namely UV-Vis and differential pulse voltammetry. A viability study was carried out with L929 cell culture.
The article fits the journal´s scope, is well organized and written in good English; however, some typo mistakes also occurred (see suggestions below).
Regarding the novelty of the article, this is not clear. The authors mentioned “numerous studies relating to the effects of the UV damage of AZA and its metabolites”, is not clear if any of them are electrochemical, nevertheless this affirmation must be supported by references. The authors should clearly highlight in the Introduction what is the novelty that their manuscript is bringing, because this aspect is missing.
The article deserves to be published, after proper revision.
Suggestions to authors:
Page 1, Abstract: Please introduce the abbreviation AZA first time when using “azathioprine” in order to justify its use afterwards.
Page 1, Abstract: Please correct “takes occurred” by “occurred” or “takes occurrence”.
Page 1, Keywords: Please correct the word “azathioprime”.
Page 1, Introduction: Please insert “.(dot)” after references [5-9].
Page 1, Introduction: Please correct “inhibit” with “inhibits” in “inhibit DNA, RNA…”
Page 2, Introduction: Please correct the name “Samaranda”.
Page 2, Introduction: Please correct “patience” with “patients” in “The high risk of skin cancer in patience…”
Page 2, Scheme 2: Please carefully verify the scheme because one of the chemical structures seems that suffered deformation.
Page 4, Viability assay: Please insert “.(dot)” at the end of the sentence.
Page 9, Figure 6: This is not the correct figure, please check and introduce the right one.
Page 13, Reference 26: Please correct “Dna” with “DNA” in the name of the article.
Author Response
The manuscript was revised according to reviewer suggestions and indications. The goal of this research paper was to reveal the effect of photodegradation of AZA on DNA using electrochemically and spectrophotometrically techniques. The novelty of this study consist both in highlighting the interaction between UV degradation products and DNA through electrochemical methods, before and after photodegradation, and in studying their effects on fibroblast cells L929 with the expectation that the result obtained will bring new insights on the influence of photodegradation of azathioprine on DNA and cells.
The article deserves to be published, after proper revision.
Suggestions to authors:
Page 1, Abstract: Please introduce the abbreviation AZA first time when using “azathioprine” in order to justify its use afterwards.
Page 1, Abstract: Please correct “takes occurred” by “occurred” or “takes occurrence”.
Page 1, Keywords: Please correct the word “azathioprime”.
Page 1, Introduction: Please insert “.(dot)” after references [5-9].
Page 1, Introduction: Please correct “inhibit” with “inhibits” in “inhibit DNA, RNA…”
Page 2, Introduction: Please correct the name “Samaranda”.
Page 2, Introduction: Please correct “patience” with “patients” in “The high risk of skin cancer in patience…”
Page 2, Scheme 2: Please carefully verify the scheme because one of the chemical structures seems that suffered deformation.
Page 4, Viability assay: Please insert “.(dot)” at the end of the sentence.
Page 9, Figure 6: This is not the correct figure, please check and introduce the right one.
Page 13, Reference 26: Please correct “Dna” with “DNA” in the name of the article.
The text was modified according to the reviewer’s suggestion.